# The Molecular Medicine PhD program alumni perceptions of career preparedness

**Valerie Chepp[1], Claire Baker[2], Sarah Kostiha[2], Jonathan D. Smith** [ID][2]*

**1** Department of Quantitative Health Sciences, Healthcare Delivery and Implementation Science Center, Cleveland Clinic, Cleveland, OH, United States of America, **2** Department of Molecular Medicine, Cleveland Clinic Lerner College of Medicine, Case Western Reserve University School of Medicine, Cleveland, OH, United States of America

* smithj4@ccf.org

**Data Availability Statement:** All relevant data are within the paper and its Supporting Information files.

**Funding:** J.D.S. received funding from the National Institutes of Health T32GM137868 (https://www.

## Abstract

Over the past two decades, graduate programs have sought to meet the rising need for cross-disciplinary biomedical and translational research training; however, among program evaluation efforts, little is known about student satisfaction with these programs. We report survey results aimed at assessing the overall satisfaction of Molecular Medicine (MolMed) PhD program graduates with their training program and subsequent employment, their research productivity since graduation, and the program elements important for entering their diverse career choices. The survey consisted of quantitative and qualitative instruments and was deployed in June 2020 via email to 45 alumni who had graduated at least two years prior. Investigators assessed mean and median Likert scale data and they conducted a qualitative content analysis on all open-ended narrative survey data using inductive analysis to identify themes. Of the 45 contacted, 26 PhD graduates of the MolMed program responded to the survey. Overall, graduates felt the MolMed curriculum prepared them well for their current career (mean 3.4 out a 4-point Likert scale); and, knowing what they know now, they would likely pursue a PhD degree again (mean 3.7 out of 4). Four overarching themes emerged from the content analysis of the narrative survey data: curriculum and other training experiences; professional skills; importance of a strong advisor/mentor; and, networking and career development. Overall, alumni were satisfied with their MolMed Program experience. They found the curriculum to be strong and relevant, and they believed that it prepared them well for their careers. There may be opportunities to embed additional skills into the curriculum, and the program should continue to offer a strong mentoring and clinical experience, as well as train students for diverse career trajectories.

## Introduction

As a nation we have made an enormous investment in the life sciences, resulting in a flood of new information and an unprecedented increase in our understanding of the fundamental processes of biology and the mechanisms of disease. Harnessing these discoveries in the service of human health requires cutting-edge scientific expertise and the ability to work closely with

nigms.nih.gov/). The funders had no role in study design, data collection and analysis, decision to publish, or preparation of the manuscript.

**Competing interests:** The authors have declared that no competing interests exist.

physicians, the latter of which is facilitated by the integration of medical knowledge into PhD training. Further, as highlighted in the National Institutes of Health (NIH) Roadmap, "the scale and complexity of today's biomedical research problems increasingly demands that scientists move beyond the confines of their own discipline and explore new organizational models for team science" [1]. This need for cross-disciplinary training provided the rationale for the Molecular Medicine (MolMed) Training Program. The MolMed Training Program overall goals are to integrate medical knowledge into PhD training and to prepare PhD students to meet these enormous challenges and become potential leaders in the next generation of translational researchers. A related objective is to prepare our PhD students to communicate effectively with physicians to foster potential collaborations during their training period and after they graduate. This approach to advancing biomedical and translational research is distinct from and complementary to dual-degree MD–PhD training programs, which seek to integrate research training into medical education [2, 3].

The MolMed PhD Program is run collaboratively at Case Western Reserve University School of Medicine (CWRU-SOM) and the Cleveland Clinic (CC). The first class of students matriculated in 2007, and classes or 6–12 students per year have matriculated since then. The Howard Hughes Medical Institute 'Med-Into-Grad' initiative supported the program during two years of planning and its first six years of operation [4]. Subsequently, the program has been supported by T32 training grants from the National Institutes of General Medicine of the National Institutes of Health, USA, and by internal funding from the CC Lerner Research Institute.

To achieve the Program goals, CWRU-SOM/CC faculty developed a new curriculum, which has been evolving since its inception. The first course the students take is human physiology, so that the students can relate all of their molecular and cellular studies back to human organ systems. The curriculum also includes a course on principals of clinical research, which encompasses clinical trial methodology, bioethics/informed consent, and advanced biostatics using R statistical software. The molecular mechanisms of human disease course combines grant writing instruction and preparation with a bedside to bench approach for a selected human disease (recent diseases chosen are type 2 diabetes, colon cancer, and multiple sclerosis). However, the central feature of the training program is that each student, in addition to selecting a thesis research advisor, also selects a clinical co-mentor in an area related to the student's research. The clinical co-mentor serves on the student's thesis committee and facilitates the independent study clinical experience course during the student's second year in the program. Often, a close interaction between the student and the clinical co-mentor continues throughout the student's time in the program, particularly when there is collaborative research between the thesis advisor and clinical-co mentor, which may culminate in obtaining clinical samples for analysis in the research lab, and even clinical trial design and performance. A description of the Mol Med training program along with comparisons to a similar program at Baylor University was published in 2013 [5].

The curriculum for the program is not static and changes have been made over the years to drop courses that the students did not value, as well as add courses and other training opportunities to meet professional training objectives. The curriculum committee, comprised of the course instructors and program director, oversees curriculum changes, which are developed in response to detailed anonymous course evaluations by the students, quarterly meetings to review the evaluations and share best practices, and an annual curriculum retreat with representatives from students in years 1 to 3 of the program.

As of January 2022, the program has matriculated 127 students, with 65 PhDs awarded with an average time to degree of ~5.5 years, and 52 students currently enrolled. We have a retention rate of 92%. Nineteen underrepresented minority (URM) students who are US citizens or

permanent residents are among the 127 matriculants, representing 15% of all entering students. Of these, 4 graduated with their PhDs, 7 are still enrolled, one dropped out in Year 1 (did not want to pursue research), and two departed after completing Year 2 with MS degrees. We have been able to track first careers for all of our graduates since program inception. Seventy percent of our graduates start off in academic or NIH/government postdoctoral positions, although others start in industry or biotech (13%), other academic positions or education (6%), clinical practice or clinical project management (6%), government regulatory (3%), and as an unexpected consequence of our medical knowledge training, 6% have entered medical school after graduation. This agrees well with recent life science PhD surveys that found that ~65% went into a postdoctoral position as their first position after graduation, with 88% of these in academia or government [6]. We have current career information for most of our PhD graduates. Currently, 47% are in academic or NIH postdoctoral or higher academic research positions, 26% are in industry or biotech, 8% are in other teaching or education, 11% are in clinical practice or clinical project management, 6% in government regulatory, 2% are in other research management positions.

The NIH has recognized the need for on-ramps for MDs into research training, including obtaining a PhD during advanced clinical training [7]. Beginning in the 2017 academic year, we recruited our first student into the PRISM track (Physician Researchers Innovating in Science and Medicine), a new track for MD clinical residents and fellows with protected time for research, to obtain a PhD in the MolMed Program. We modelled this program after the similar STAR Program at UCLA, which has had excellent outcomes [8]. Five years after implementing this track, we have matriculated 11 PRISM students.

We sent a survey in the summer of 2020 to 45 graduates, who had graduated at least two years prior, in order to gauge student satisfaction with career preparedness and the program curriculum. Here we provide some more details of our program curriculum and the results of this survey.

## MolMed curriculum

The outline of the current courses for the MolMed program is shown in Fig 1. Students start in mid-July with the individual courses described in S1 File. Alumni responses may not reflect the current curriculum; although, many of the courses have been offered every year (albeit with some content changes), including the Clinical Experience, Physiology, Cell Biology, Metabolism, Molecular Bio/Gene Regulation, Immunology, Genetics/Bioinformatics, and Principles of Clinical and Translational Research. Details of our course offerings, other student requirements, trainee oversight, and professional and career development opportunities are found in S1 File.

| Year 1 | | | Year 2 | | |
| --- | --- | --- | --- | --- | --- |
| **Summer** | **Fall** | **Spring** | **Summer** | **Fall** | **Spring** |
| Tools Course (with Lab)<br><br>Human Physiology and Disease | Cell Biology<br><br>Metabolism | Molecular Bio/Gene Regulation<br><br>Immunology<br><br>Genetics/Bioinformatics | Principles of Clinical and Translational Research | Molecular Mechanisms of Human Disease (w/grant writing) | Independent Study Clinical Experience |
| Trainer Poster Session and Presentations | Student Seminar Series | Student Seminar Series | Thesis Committee and Clinical Mentor chosen<br><br>Student Seminar Series | First Thesis Committee Meeting<br><br>Student Seminar Series | Qualifying Exam<br><br>Student Seminar Series |
| Lab Rotation 1 | Lab Rotation 2 | Lab Rotation 3 | Thesis Research | Thesis Research | Thesis Research |

**Fig 1. First two years of the current MolMed curriculum.**

## Method

### Alumni survey design

We started with two surveys from Cornell University Graduate School: the Doctoral Career Outcomes Survey and the Survey of Earned Doctorates [9]. We combined parts of these two surveys and modified them to fit our Program. The final survey is shown in S2 File. We obtained IRB approval (protocol 07–542, REGISTRY: Program Evaluation of Molecular Medicine PhD Program) for deploying this survey to our alumni and the subsequent analysis. The survey cover letter was sent in June 2020 via an email to 45 alumni who had graduated at least two years prior. The complete email text is shown in S3 File and the raw survey response data is shown in S4 File.

### Statistical analysis

We assessed mean and median Likert scale data using Excel. We prepared figures using GraphPad Prism v9.0 software.

### Qualitative analysis

We conducted a qualitative content analysis on all open-ended narrative survey data. We used Excel to sort and organize qualitative data, and we conducted analysis in two stages, using inductive analysis and guided by a ground theory approach [10]. First, we analyzed all narrative data through an inductive process of open coding. Next, we identified and further refined emerging concepts and categories through focused coding. Direct quotes from respondents were in some cases subject to minor editing for grammar and clarity, such as spelling out words abbreviated by the respondent.

## Results

### Survey respondents and post-graduation outcomes

26 PhD graduates of the MolMed program, out of the 45 contacted, responded to our survey; although, not all respondents answered all questions. The range and median dates of entry into our program and graduation are shown in Table 1. Notably, with a matriculation range of six years, this survey captures alumni perceptions of the program when it was still quite new. The inferred time to degree ranged from 2.8 to 6.8 years with a mean and median of 5.3 and 5.5 years, respectively.

Of the 20 respondents who answered the question about their first job after graduation, 18 responses indicated postdoctoral research positions, 1 response indicated research staff, and 1 response indicated a medicine residency. The employment sector for the first postgraduate job was overwhelmingly academic postdoctoral (15 responses) with 1 response each for industry/ biotech, education, and hospital. The duration in years for the first postgraduate job ranged from 1 to 5 years, with a median of 2 years (Table 1).

**Table 1. Demographics of alumni respondents.**

|  | Minimum | Maximum | Mean | Median | # of Responses |
|---|---|---|---|---|---|
| Matriculation date | July 2007 | July 2013 |  | July 2009 | 26 |
| Graduation date | May 2011 | Aug. 2018 |  | Dec. 2015 | 26 |
| Inferred years to degree | 2.8 | 6.8 | 5.3 | 5.5 | 26 |
| # of Years in first job | 1 | 5 | 2.44 | 2 | 18 |
| # of Years in current job | 1 | 4 | 2.12 | 2 | 26 |

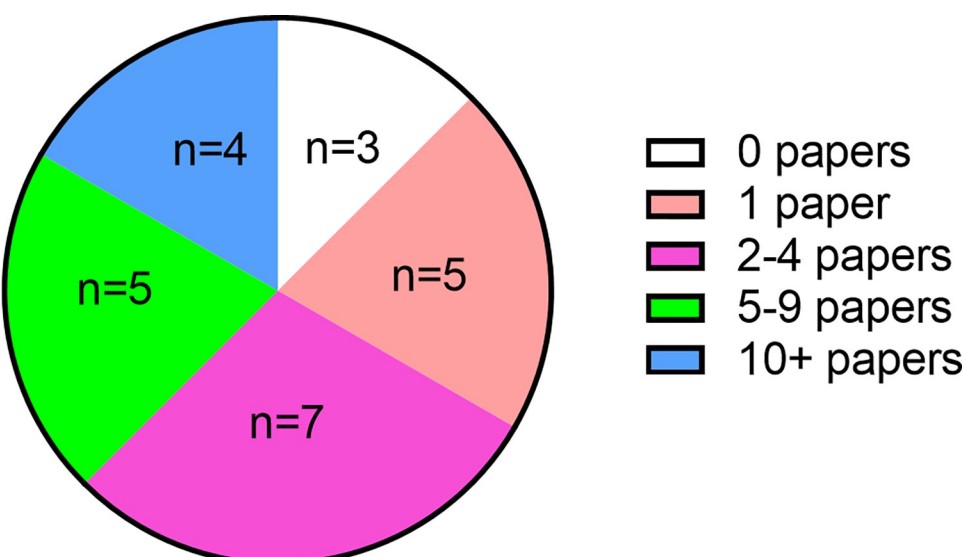

**Fig 2. The number of research publications since graduation.** The values shown were options in a pull down list.

We received 26 responses to current job position. We grouped related job descriptions, yielding 16 in postdoctoral, research assistant, or research scientist positions; 2 in medicine residency; 1 in a regulatory agency; 4 in industry or pharmaceutical companies; and, 3 in Assistant Professor positions. The duration in years for the current position ranged from 1 to 4, with a median of 2 years (Table 1). We asked about satisfaction with their current position, and provided 4 potential responses: very dissatisfied, dissatisfied, satisfied, and very satisfied. Of the 23 respondents, 3 were very dissatisfied, 7 were satisfied, and 13 were very satisfied. On a 4-point Likert scale the mean was 3.3 and the median was 4.

To determine if our graduates are continuing to be productive researchers, we asked them how many research papers they have published since graduation, giving the options of: 0, 1, 2 to 4, 5 to 9, and 10 or more. There were 24 respondents, with the median response being 2 to 4 papers. The breakdown of the responses is shown in Fig 2, with 9 respondents (37.5%) in the two highest groups having at least 5 research publications.

We asked about post graduate research grants. Of the 25 respondents, 11 had received neither federal nor private research grants, 6 had received only private research grants, 4 received only federal research grants, and 4 had received both federal and private research grants (Fig 3). Summing the latter two groups yields that 8 of 25 (32%) respondents had received federal research grant support.

In order to gauge alumni overall satisfaction with the program and the pursuit of a PhD, we asked the buyer's remorse question, "Given the perspective you have gained since completing the Molecular Medicine PhD program, if you could start again what would you do?" The respondents chose between: definitely not, probably not, probably, and definitely. No one selected "definitely not", and only one selected "probably not". The majority selected "definitely". The full breakdown is in Fig 4.

## Program evaluation by graduates

The narrative survey data centered on graduate satisfaction with the MolMed training program. These data were grouped into nine analytic domains; seven domains consisted of a single, stand-alone survey question and two were comprised of multiple questions on a related

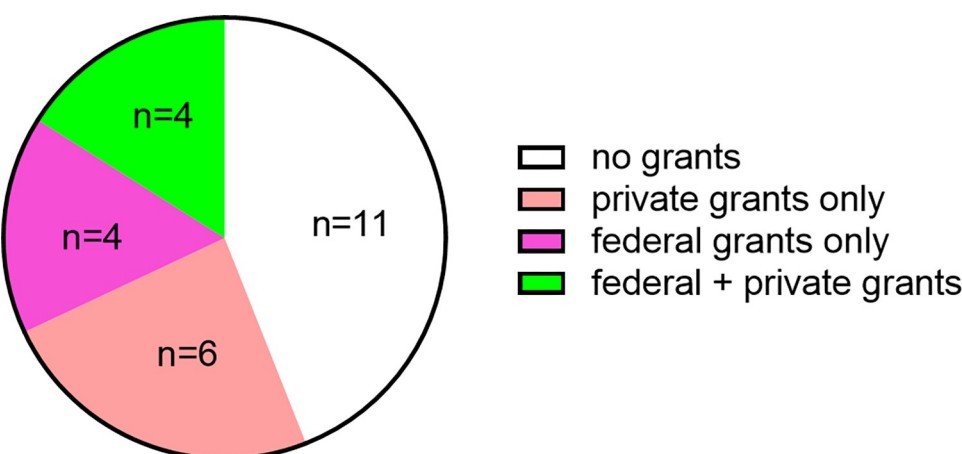

**Fig 3. Research grants received since graduation.** Respondents could click yes or no for private and federal research grants.

topic. Four themes emerged from our analysis of the narrative survey data: 1) curriculum and other training experiences; 2) professional skills; 3) importance of a strong advisor/mentor; and, 4) networking, career development, and job placement. Below, we present these thematic findings and, where applicable, place them in context with our quantitative results.

**1. Curriculum and other training experiences.** Alumni perceive their MolMed PhD training to be broad and comprehensive, and they see this as one of the program's strengths. This sentiment is exemplified by an alumnus working as an industry group leader in biotech/pharma, who said: "I draw from the knowledge and experience daily, partly because of how broad and comprehensive the program curriculum is. This lets me understand all areas of clinical/translational research, not just one."

Overall, alumni felt that the MolMed PhD classroom curriculum prepared them well for their respective career fields, with a mean and median 4-point Likert scale responses of 3.40 and 3, respectively (Table 2).

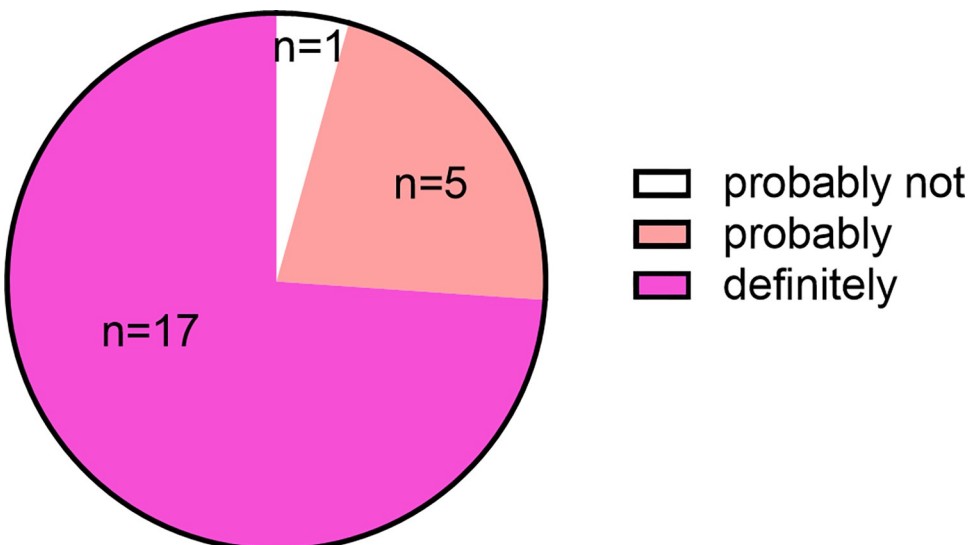

**Fig 4. Respondent's answers to whether they would pursue a PhD again knowing what they know now.** On a 4 point Likert scale, the mean was 3.70 and the median was 4.

**Table 2. Alumni preparedness.**

| Mean | Median | # of Responses |
|---|---|---|
| 3.40 | 3 | 25 |

How prepared did you feel entering your career field based on the Molecular Medicine PhD classroom curriculum? 1 = Not prepared at all; 2 = Not very prepared; 3 = Prepared; 4 = Very Prepared

When asked which factors were significant in obtaining their current positions, the most cited factors were: 1) perceived quality of their academic work; 2) reputation of CWRU and Cleveland Clinic; 3) the clinical experience and focus on translational research; and 4) thesis advisor/committee, faculty, and program administration (Table 3).

When given the opportunity to expand upon specific classes/topics/curriculum that contributed to their preparedness, they cite the integration and totality of the curriculum. For instance, one alumnus who works as a scientist in biotech/pharma and who felt "very prepared" by the MolMed classroom curriculum said, "I think the multidisciplinary nature of our program pushed us to understand a wide-range of topics and fundamentals." Another alumnus who felt "very prepared" and who works as a postdoctoral fellow in an academic medical center also referenced the totality of the curriculum, citing "All the classes we took as well as the training we received in our respective labs." Similarly, an industry scientist working in biotech/pharma who felt "very prepared" entering their field said, "Each class provided something different. I worked on novel cancer therapy in grad school and transitioned to immuno-oncology in industry. Immunology helped but the broad curriculum prepared me to adapt."

Other alumni echoed this perception that the comprehensive nature of the MolMed PhD training contributes to an ability to adapt to different work environments. A senior scientist working in biotech/pharma who "strongly agreed" that the program's content supported their research and/or professional goals stated, "The program exposed me to a wide range of skill sets that led me to understand and be able apply my knowledge widely with an ability to be flexible and adjust to a dynamically changing work setting."

While the broad and comprehensive nature of the MolMed PhD program was highlighted as an asset by several alumni, others made reference to specific program elements. For example, some alumni identified particular courses as especially contributing to their career preparedness, with the most cited courses being immunology, genetics, and nucleic acids/gene expression.

The clinical experiences embedded in the curriculum was repeatedly highlighted by alumni as a specific program element that contributed positively to their training. This finding is

**Table 3. Significant factors.**

| | Count |
|---|---|
| Career readiness programs | 3 |
| Clinical Experience/focus on translational research | 15 |
| Connections and networking with CWRU/ CC program alumni | 6 |
| Perceived quality of my academic work | 17 |
| Reputation of Case Western Reserve University and Cleveland Clinic | 16 |
| Reputation of Molecular Medicine Program | 5 |
| Support and activity of thesis advisor, thesis committee, other CC/CWRU faculty, or program administration | 13 |
| Other significant factors | 5 |

Which of the following were significant factors in helping you to land your current position? (Mark all that apply)

supported by the quantitative survey data, as well (Table 3). For example, one alumnus who reported feeling "very prepared" by the curriculum for their career as an Assistant Professor at a university felt as though the clinical experience is what prepared them most: "The emphasis on the clinical connection with everything we learned was probably the most important factor rather than any single class." Another alumnus who works as a postdoctoral scholar at a university stated that "Easy access to clinical environment and training from field experts" was an important competency for doctoral students entering their field. While many respondents reported having positive clinical experiences, one respondent, a research scientist working in biotech/pharma, did not, stating "I think the curriculum has evolved since my days. I hope the clinical experience is better. I did not have a lot of guidance or support there. When I shadowed MDs I think they thought I was a medical student."

While alumni highlighted the MolMed curriculum as being important for their own career preparedness, they identified other elements of the MolMed PhD training—beyond the classroom curriculum—as being more important for doctoral students interested in entering their career field. Specifically, the quantitative data show that alumni viewed experience gleaned through collaborative work with lab students/personnel (mean score 3.68), work with their thesis advisor/committee (mean score 3.63), opportunities to practice research methods to their field (mean score 3.68), and presenting work at a professional conference/seminar (mean score 3.80) as all being more important than the classroom curriculum (mean score 3.36) for others entering into their respective career fields (Table 4). Although exposure to non-academic or non-research careers was important (mean score 3.16), it was the least valued of all of the elements queried.

**2. Professional skills.** Specific professional skills and competencies emerged as a second prominent theme from the data, which respondents felt were either acquired or not acquired from their MolMed PhD training experience. Our survey asked alumni the importance of various competencies/skills related to their current field of work (Table 5). The median scores were 3 or 4 for all of these competencies. One finding of interest is that competency in ethics and integrity was rated the highest of overall importance (mean score 3.64). Distinct from rating the importance of these competencies, we also asked how satisfied the alumni were with the training provided by the MolMed program in these competencies (Table 6). The median scores were all 3, again with ethics and integrity getting the highest marks (mean score 3.43). Within this professional skills theme, two subthemes emerged: (a) applied, hands-on job skills and (b) social skills.

*2a. Applied, hands-on job skills.* Throughout the survey, alumni identified specific hands-on job skills important in their training and respective career fields. When asked what advice

**Table 4. Importance of training elements.**

|  | Mean | Median | # of Responses |
|---|---|---|---|
| Experience gained through Molecular Medicine PhD classroom curriculum | 3.36 | 3 | 25 |
| Experience working collaboratively with students and other lab personnel | 3.68 | 4 | 25 |
| Experience working with your thesis advisor and committee | 3.63 | 4 | 24 |
| Exposure to careers outside of academic scholarship/research | 3.16 | 3 | 25 |
| Practice of research methods in your field | 3.68 | 4 | 25 |
| Presentation of work at a professional conference and/or seminar | 3.80 | 4 | 25 |

For doctoral students interested in your career field (broadly defined), how important are the following elements of your Molecular Medicine PhD training? 1 = Not important at all; 2 = Not very important; 3 = Important; 4 = Very Important

**Table 5. Importance of competencies.**

|  | Mean | Median | # of Responses |
|---|---|---|---|
| Academic and professional writing | 3.48 | 4 | 23 |
| Bench-based wet lab research | 3.45 | 4 | 22 |
| Ethics and Integrity | 3.64 | 4 | 22 |
| Genetics/Bioinformatics | 2.95 | 3 | 22 |
| Human Physiology and Disease | 3.22 | 3 | 23 |
| Medical knowledge including the Clinical Experience course | 3.26 | 4 | 23 |
| Molecular/Cellular Biology | 3.57 | 4 | 23 |
| Statistics/Epidemiology | 3.26 | 3 | 23 |
| Other Competencies | 3.14 | 3 | 21 |

How important are each of the following competencies for doctoral students entering your field? 1 = Not important at all; 2 = Not very important; 3 = Important; 4 = Very Important

they'd give to an incoming student, a university post-doctoral research fellow advised: "Try to learn as much as you can during your rotations. The more techniques you know the more marketable you are." This advice was reiterated by another alumnus, an academic postdoc, who identified "Technique specific knowledge" as being an important competency for doctoral students entering their field.

Presentation skills were often mentioned by alumni as being an important proficiency for their career fields; this finding emerged in the quantitative data analysis as well (Table 4). When asked to "list other competencies" important for doctoral students entering their field (beyond those listed in Table 5), of the 18 who responded, one-third (n = 6) listed presentation skills as an important competency. Asked what advice they'd give to incoming doctoral students, a university postdoctoral scholar said, "Attend as many seminars as possible, they will help you to develop presentation skills and simultaneously train you in scientific thinking and understanding." Business, grant, and project/budget management skills were also listed as important hands-on job skills by two alumni. Another alumnus, a regulatory specialist in a hospital, suggested offering "a writing course" that would develop diverse writing skills, "with modules for publications, grants, and posters, etc."

Computer/technological skills, specifically, were emphasized by several respondents as being valuable hands-on skills for their training and careers. When asked to provide a

**Table 6. Satisfaction with competency training.**

|  | Mean | Median | # of Responses |
|---|---|---|---|
| Academic and professional writing | 3.04 | 3 | 23 |
| Bench-based wet lab research | 3.43 | 3 | 23 |
| Ethics and Integrity | 3.43 | 3 | 23 |
| Genetics/Bioinformatics | 3.04 | 3 | 23 |
| Human Physiology and Disease | 3.39 | 3 | 23 |
| Medical knowledge including the Clinical Experience course | 3.35 | 3 | 23 |
| Molecular/Cellular Biology | 3.35 | 3 | 23 |
| Statistics/Epidemiology | 2.91 | 3 | 23 |
| Other Competencies | 2.94 | 3 | 18 |

How satisfied were you with the Molecular Medicine PhD curriculum and training in these competencies? 1 = Not important at all; 2 = Not very important; 3 = Important; 4 = Very Important

suggestion on how the Molecular Medicine PhD curriculum could better align with important competencies in their field, a data scientist working in business/industry said, "I know things have changed since I was doing coursework 10 years ago, but I hope that today's students are learning some coding basics as a requirement of the program. Strongly recommend R or Python." Throughout the survey, this alumnus repeated how important learning to code has been for a successful career in their field. Elsewhere in the survey they said, "Ability to write readable code in either R or Python, and to use good research reproducibility tools in bioinformatics. This was a critical step for me in succeeding in human genetics research." As stated in the MolMed curriculum (S1 File), our current courses do teach how to use R software and we are planning to add additional training in computational biology including an introduction to R coding.

An assistant professor at a college/university advised incoming doctoral students to "learn bioinformatics and computational biology approaches." Similarly, an industry group leader in biotech/pharma said that "There should be a dedicated and through bioinformatics and programming component. The Stanford Immunology PhD, for example, teaches every single student the Python programming language" when suggesting how the Molecular Medicine PhD curriculum could better align with competencies in their field.

*2b*. *Social skills*. Alumni also listed many social skills as being important competencies for successful careers in their fields. Two of the 18 alumni who listed "other competencies" important for doctoral students entering their field used the term "soft skills," and many others elaborated citing communication skills (n = 4), collaboration skills (n = 3), and negotiation skills (n = 2). Notably, as previously mentioned, alumni rated the social skill of competency in ethics and integrity as the highest overall important competency for doctoral students entering their field.

**3. Importance of strong advisors/mentors.** A third prominent theme appearing throughout the qualitative survey data focused on the importance of a strong advisor/mentor. Table 3 shows that many of the respondents thought that support from the thesis advisor and committee/program was important for them obtaining their job. When asked to provide a supporting reason for the extent to which they felt the MolMed PhD program supported their research and/or professional goals, 2 of 10 alumni mentioned their advisor/mentor. A postdoctoral research fellow who "strongly agreed" that the program supported their research/professional goals said, "The translational nature of the program, my clinical mentor and the collaboration possible at the Cleveland Clinic prepared me for my career." A scientist working in biotech/pharma also "strongly agreed" that the program supported their goals and said, "I was so fortunate to have an excellent thesis advisor that pushed me to think outside the box while still encouraging project progression towards graduation."

While some alumni reported having a positive advising/mentoring experience, which contributed to their satisfaction with the program, others had a less positive experience and/or saw this as an opportunity where the program could improve. Offering a suggestion for how the curriculum could better align with competencies important in their field, a regulatory specialist working in a hospital suggested including opportunities for "Mentorship outside of the lab."

This theme around mentoring and advising also showed up in much of the advice alumni would give incoming doctoral students. A research scientist working in biotech/pharma said, "Pick the advisor you click with the most. More important than the work you do is the connection and support you have with and from him/her." Another scientist working in biotech/pharma said, "Pick the lab because of the principal investigator, with the subject area secondary." An industry group leader working in biotech/pharma said, "Take lab rotations and advisor selection very seriously."

The importance of an advisor/mentor is also evident in the "additional information" this industry group leader chose to communicate to the MolMed PhD program: "It would be nice to have transparency about the track record of any prospective advisor when selecting during rotations. For example, how many students has the advisor mentored and what are the trainee's feedback and outcomes. Selection of advisor is so important and I felt like I was making decisions off perception and hearsay/rumors rather than having objective information about advisor performance and student feedback."

**4. Networking, career development, and job placement.** A fourth theme emerging from the qualitative analysis focused on networking, career development, and job placement. Alumni frequently mentioned "networking" in their narrative responses throughout the survey. Notably, this discussion didn't focus on networking as a skill, but rather on the importance of networking for career success. For example, one alumnus, an industry group leader who works in biotech/pharma, attributed "Networking at national conferences with large industry presence" as a significant factor that helped them land their current position. Another alumnus, a research scientist working in biotech/pharma also credited "networking" to getting hired in their current role, stating "Someone from my postdoc knows my now boss." Alumni also noted the importance of networking when imparting words of advice to incoming doctoral students. For example, a medical science liaison working in biotech/pharma said, "Learn as many skills as possible from experts in different labs, or even departments. Join professional societies to grow your network and learn about future career opportunities. Attend conferences." Another alumnus, a neurosurgery resident, advised incoming students with the simple statement: "Networking is incredibly important." Only one alumnus, a postdoctoral research fellow, listed networking as a competency, that is, a skill to be learned. This alumnus cited networking as an important competency for doctoral students entering their career field.

Career development and job placement emerged as additional themes. In general, alumni felt prepared and satisfied with the career training they received. This sentiment was expressed by one alumnus, an academic postdoctoral scholar: "The exposure to equipment and field experts in the Lerner Research Institute are unique and strongly help to set up the basis for future training/working in research." This finding from the qualitative data aligns with those from the quantitative data; that is, overall, alumni felt the MolMed PhD classroom curriculum prepared them well for their respective career fields (Table 2) and they were satisfied with how the MolMed Program trained them in various competencies important for their fields (Table 6).

The qualitative data, however, reveal two additional subthemes within this overarching theme: (1) opportunities for additional career development and, relatedly, (2) desire for more training for diverse career trajectories, including non-academic careers.

First, alumni identify additional career development opportunities not present in the curriculum (at the time of their tenure). This subtheme overlaps with some previously presented findings in the "professional skills" and "advisor/mentor" theme. Here, we present additional content mentioned by alumni, which they directly connect to their career development. For example, a data scientist working in business/industry said, "I love my work as a data scientist, and hope that data science is included in the career development seminars these days." A scientist working in biotech/pharma said, "I would encourage the program to offer more guidance on choosing a post-doc lab or career post-graduation. With graduation always feeling so far away, it was impossible to anticipate that you were a year out to start communicating with principal investigators. Furthermore, students need to know how to evaluate a principal investigator for success—my postdoctoral experience was absolute garbage compared to my graduate training and I wish I knew more about how to evaluate this advisor prior to joining a postdoc lab."

Alumni also expressed a desire to be trained in more diverse career trajectories, including those beyond academia. A regulatory specialist in a hospital said, "The program was geared toward publications and thesis only and did not supply support for career training, career options, or scientific writing." A postdoctoral fellow in an academic medical center felt the MolMed Program "mostly supported careers in academia in terms of the how the exams were given and what the tasks consisted of." This desire for more career training may explain why only 3 of 24 respondents cited "career readiness programs" as being a significant factor in helping them secure their current position (Table 3).

When alumni were asked if they would pursue the PhD degree knowing what they know now, only one alumnus said they would "probably not" pursue a doctoral degree again (Fig 4). This respondent cited career development and job placement issues as their reason why, stating that it's "Very hard to get out of post-doctoral positions with very little opportunity for advancement and no direct path to industry positions."

## Discussion

Overall, alumni perceived the MolMed Program in a positive light. They found the program to offer a strong and relevant curriculum. They perceived the clinical experiences embedded in the curriculum, along with the close mentoring they received, to be particular program strengths. Overall, alumni felt that the MolMed curriculum, training, and mentoring has prepared them well for their careers.

While alumni were generally satisfied with their MolMed training, looking ahead, our analysis spotlights opportunities for further reflection and potential programmatic modification. One area for reflection might be to look at how, or to what extent, we teach various skills in the curriculum. For example, while we currently offer extensive training in grant writing, we might reexamine how we teach other forms of scientific writing, and for different audiences. We might also consider intentionally embedding networking skill-building opportunities into the curriculum, particularly given the importance of networking to alumni's career success [11]. Our growing number of program graduates and expanding alumni base may offer additional networking opportunities for current students and recent alumni. Programmatic self-reflection and a willingness to make modifications is important to maintaining a strong and relevant curriculum. Our decision to start teaching R software in our current courses is an example of our ongoing efforts to introduce new skills into our curriculum. The fact that some alumni wanted more training on bioinformatics and computational approaches even though such competencies were reported in the survey as the least relevant for success is not surprising based on the diversity of career path outcomes for our alumni, not to mention most do not pursue careers in computational biology. Our alumni have different needs and different careers trajectories.

A responsive curriculum that trains students in a wide array of skills helps to support diverse career trajectories in the biomedical field [12]. A graduate curriculum that prepares doctoral students for multiple career options is a recognized need within the field of biomedical and health sciences, as available tenure-track academic positions continue to decline [13] and students express an interest—and desire to be trained—in various careers, both within and outside of traditional academic pathways [12, 14–16]. The MolMed Program acknowledges and understands these shifting economic, job market, and doctoral workforce realities. The executive and curriculum committees, along with the course evaluation instruments, have promoted the reflective nature of the MolMed training program, with continual course and extracurricular refinements. The continuation of our alumni surveys along with tracking the career outcomes of our graduates are two efforts that further support curricular reflection and

refinement, and are recommended practices within the field of biomedical program evaluation [17].

Finally, our analysis underscores that we must continue to ensure that all students have a strong mentoring experience. Not only is this a signature feature of our program, but alumni point to the importance of good mentoring for their career success and satisfaction with the MolMed Training Program. In this regard, our alumni are similar to those in other biomedical programs who emphasize the importance of strong mentorship [15, 18]. While most alumni had a positive mentoring experience, some felt it could be improved, such as by making it more transparent. Efforts to make the mentoring experience more transparent and robust are in place. As the academic advisor for all first-year students, the program director meets individually with first-year students to help guide them through the mentoring selection process for their rotations. As part of this advising, the program director shares information about past performances of mentors under consideration. This discussion becomes especially important when students select their thesis mentor. As the program has matured, more knowledge has been gleaned and program director guidance has become more informed. Additionally, all new mentors must go through the University of Minnesota's online training program, which requires participants to write a mentoring plan, which must be reviewed and approved (https://ctsi.umn.edu/training/mentors/mentor-training).

The current study captures alumni perceptions of the MolMed Program in its early years, before significant programmatic revisions occurred. Future alumni surveys will be better positioned for longitudinal analysis. For example, subsequent surveys will be able to account for alumni perceptions by matriculation date, and how program improvements over time might shape perceptions. The early matriculation dates for these survey respondents might also explain why the reputation of the school was three times more important in helping respondents secure their current position. For these early program participants, the program had not much time to yet earn a reputation. The ability for future surveys to collect data from alumni who represent a wider range of matriculation dates will allow us to evaluate the career significance of the program's reputation over time.

One of the strengths of the current study is the incorporation of both quantitative and qualitative questions in the survey, allowing us to gain more insight into the program strengths and targeted areas for improvement. One limitation of our study is the selection bias inherent in obtaining responses from 26 of the 45 alumni that we contacted. It is possible that the alumni who responded are more apt to have positive training experiences compared to those who did not respond. Even though the median time since graduation of our respondents was only ~4.5 years, we were pleasantly surprised that ~32% had already received federal research support after graduation, although we did not distinguish between individual research project awards (R01 equivalent) and training and development awards (F32 equivalent). Finding comparative data on research funding is challenging; however, some evidence suggests that our program outcome on this measure is encouraging when compared to MD-PhD program graduates, where 22% of women and 34% of men report being a principal investigator on a current NIH Research Project Grant [19]. Although not a direct comparison since these data on MD-PhD graduates include older cohorts than ours and focus only on NIH research grants (and do not include fellowship awards), these measures offer some context for the success of MolMed alums in securing research funding after graduation.

This research may benefit others who are considering developing similar programs or those interested in conducting similar programmatic assessments. Our findings may also suggest curricular and programmatic updates for graduate training programs for biomedical and translational researchers. While the development of such programs has been critical in addressing the gap identified nearly 20 years ago in the NIH Roadmap, we must also evaluate

the effect such programs have on graduates' career trajectories. Our research on MolMed alumni's satisfaction with their training and subsequent employment, their research productivity since graduation, and their perceptions of career preparedness contributes to this larger endeavor.

## Supporting information

**S1 File. Course titles and descriptions.**
(PDF)

**S2 File. Molecular Medicine post-graduation program review and outcomes survey.**
(PDF)

**S3 File. Survey cover letter.**
(PDF)

**S4 File. Raw survey responses from each respondent.**
(XLSX)

## Acknowledgments

We thank Dr. Beth Bierer for reviewing our survey and suggesting refinements before we sent it out.

## Author Contributions

**Conceptualization:** Sarah Kostiha, Jonathan D. Smith.

**Data curation:** Valerie Chepp, Claire Baker, Sarah Kostiha, Jonathan D. Smith.

**Formal analysis:** Valerie Chepp, Jonathan D. Smith.

**Funding acquisition:** Jonathan D. Smith.

**Investigation:** Valerie Chepp, Claire Baker, Sarah Kostiha, Jonathan D. Smith.

**Methodology:** Jonathan D. Smith.

**Project administration:** Jonathan D. Smith.

**Resources:** Jonathan D. Smith.

**Supervision:** Jonathan D. Smith.

**Visualization:** Claire Baker, Jonathan D. Smith.

**Writing – original draft:** Valerie Chepp, Jonathan D. Smith.

**Writing – review & editing:** Valerie Chepp, Jonathan D. Smith.

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
