## [Decision Letter · Decision Letter 0]

8 Sep 2022

PONE-D-22-21391The Molecular Medicine PhD Program Alumni Perceptions of Career PreparednessPLOS ONE

Dear Dr. Smith,

Thank you for submitting your manuscript to PLOS ONE. After careful consideration, we feel that it has merit but does not fully meet PLOS ONE’s publication criteria as it currently stands. Therefore, we invite you to submit a revised version of the manuscript that addresses the points raised during the review process.

We look forward to receiving your revised manuscript.

Kind regards,

Maria E. Solesio Torregrosa, PharmD/PhD

Academic Editor

PLOS ONE

Additional Editor Comments :

Please, see comments from the reviewer. Thanks

Reviewers' comments:

Reviewer's Responses to Questions

**Comments to the Author**

1. Is the manuscript technically sound, and do the data support the conclusions?

Reviewer #1: Partly

2. Has the statistical analysis been performed appropriately and rigorously? 

Reviewer #1: Yes

3. Have the authors made all data underlying the findings in their manuscript fully available?

Reviewer #1: No

4. Is the manuscript presented in an intelligible fashion and written in standard English?

Reviewer #1: Yes

5. Review Comments to the Author

Reviewer #1: The manuscript "The Molecular Medicine PhD Program Alumni Perceptions of Career Preparedness" by Chepp et al. presents an analysis of the MolMed program evaluations made by alumni. While the the questions posed in the survey were appropriate, the number of responses was slightly higher than 50% of the contacted alumni and, according to what reported by the authors, they joined the program at different stages of its maturation. This particular point is not discussed and I think it should be included in order to take into account the eventual improvements that have been made based on the answers of alumni that were offered different courses. Moreover, despite the multiple choice answers are available to the reviewer in the tables, there is no access to the open ended answers which results in a lack of data made available for the reviewer to assess the analysis.

More specific comments:

1) The results reported in table 3 indicate how the reputation of the school was 3 times more important than the program itself. The authors should discuss this huge weakness.

2) Table 5 and 6 report the same question but with different results. Please correct and adjust the text accordingly.

3) Despite some alumni think the program should offer a more thorough training on bioinformatics and computational approaches the survey showed that such competency was the least relevant for success. The authors should include this discrepancy in their discussion.

4) Mentorship: after reporting some open ended answers in section 3., the authors don't discuss the negative criticism received by certain alumni on this topic. This should be added in the discussion part for a proper analysis.

6. PLOS authors have the option to publish the peer review history of their article (what does this mean?). If published, this will include your full peer review and any attached files.

Reviewer #1: No

---

## [Author Response · Author response to Decision Letter 0]

23 Sep 2022

We thank you for the opportunity to resubmit our article “The Molecular Medicine PhD Program Alumni Perceptions of Career Preparedness.” We thank the reviewer for their thoughtful comments, which we have addressed in the table below. We believe their comments have made the manuscript even stronger, and look forward to having our paper considered for publication.

1. Is the manuscript technically sound, and do the data support the conclusions?

Reviewer #1: Partly 

Response: We have revised the manuscript in order to strengthen our conclusions based on the data we present (see below). We are especially appreciative of how the Reviewer’s comments contributed to a richer and more thorough Discussion section. Thank you.

2. Has the statistical analysis been performed appropriately and rigorously?

Reviewer #1: Yes 

Response: Thank you.

3. Have the authors made all data underlying the findings in their manuscript fully available?

Reviewer #1: No 

Response: We have uploaded the raw responses for the qualitative survey questions as Supplemental File 4.

4. Is the manuscript presented in an intelligible fashion and written in standard English?

Reviewer #1: Yes 

Response: Thank you.

5. Review Comments to the Author: 

1) While the the questions posed in the survey were appropriate, the number of responses was slightly higher than 50% of the contacted alumni and, according to what reported by the authors, they joined the program at different stages of its maturation. This particular point is not discussed and I think it should be included in order to take into account the eventual improvements that have been made based on the answers of alumni that were offered different courses. 

Response: Thank you for raising this thoughtful point about survey respondents entering the program at different stages of program maturation, and potential implications for eventual program improvements. Given the survey respondents’ narrow range of matriculation date (2007-2013), for these students, the program was still in its early stages and didn’t change significantly in those first few years. Moreover, with such a small number of respondents, we didn’t feel powered to see any longitudinal effects. For example, we considered dividing the sample in half by “early matriculators” and “later matriculators”; however, each category would consist of only a 3-year range and we didn’t feel this would produce meaningful results. To address the Reviewer’s comment, we made several revisions to the manuscript to clarify and elaborate upon their point. First, when describing respondents’ demographics in Table 1, we added a sentence to underscore the limited matriculation date range of 6 years and how the survey data captures alumni perceptions of the program when it was still quite new (p. 6). Next, in the discussion section, we added a sentence reiterating the narrow range of matriculation date and the limitations this posed for a longitudinal analysis. We highlight this as an opportunity for future analysis (p. 18).

2) Moreover, despite the multiple choice answers are available to the reviewer in the tables, there is no access to the open ended answers which results in a lack of data made available for the reviewer to assess the analysis.

Response: Thank you for drawing our attention to this. We have uploaded the raw responses for the survey questions, including the open-ended answers, in Supplemental File 4. In the Methods section, we added a statement indicating that the raw data is available in Supplemental File 4 (p. 5). 

3) The results reported in table 3 indicate how the reputation of the school was 3 times more important than the program itself. The authors should discuss this huge weakness. Thank you for highlighting this finding and suggesting that we elaborate upon it.

Response: In the Discussion section, we offer a possible interpretation by building upon the fact that the program was still in its infancy for these survey respondents, and thus had yet to secure much of a reputation. We highlight this as an opportunity for future analysis (p. 18).

4) Table 5 and 6 report the same question but with different results. Please correct and adjust the text accordingly. 

Response: Thank you for catching our error! We deeply appreciate your close read of the manuscript. On page 12, we have revised the manuscript by adding the correct title of Table 6. On p. 11, we offer additional clarification on the distinction between Table 5 and 6 by adding the (boldface) phrase: “Distinct from rating the importance of these competencies, we also asked how satisfied the alumni were with the training provided by the MolMed program in these competencies (Table 6).”

5) Despite some alumni think the program should offer a more thorough training on bioinformatics and computational approaches the survey showed that such competency was the least relevant for success. The authors should include this discrepancy in their discussion. Thank you for highlighting this finding and suggesting that we elaborate upon it.

Response: In the Discussion, we underscore this discrepancy. We suggest that it unsurprising given the diversity of career path outcomes for our alumni, not to mention most of our alumni do not pursue careers in computational biology (p. 17).

6) Mentorship: after reporting some open ended answers in section 3., the authors don't discuss the negative criticism received by certain alumni on this topic. This should be added in the discussion part for a proper analysis. 

Response: Thank you for pointing out the importance of discussing the negative criticism on mentorship for a proper analysis. In the Discussion, we address the fact that some alumni felt improvements could be made to the mentoring experience, such as making it more transparent. We elaborate on efforts to make the mentoring selection process more transparent, and required training for all mentors (p. 17-18)

---

## [Editor Report · Decision Letter 1]

27 Sep 2022

The Molecular Medicine PhD Program Alumni Perceptions of Career Preparedness

PONE-D-22-21391R1

Dear Dr. Smith,

We’re pleased to inform you that your manuscript has been judged scientifically suitable for publication and will be formally accepted for publication once it meets all outstanding technical requirements.

Kind regards,

Maria E. Solesio Torregrosa, PharmD/PhD

Academic Editor

PLOS ONE
---

## [Editor Report · Acceptance letter]

9 Nov 2022

PONE-D-22-21391R1 

The Molecular Medicine PhD Program Alumni Perceptions of Career Preparedness 

Dear Dr. Smith:

I'm pleased to inform you that your manuscript has been deemed suitable for publication in PLOS ONE. Congratulations! Your manuscript is now with our production department. 

Kind regards, 

on behalf of

Dr. Maria E. Solesio Torregrosa 

Academic Editor

PLOS ONE